# A Dynamic Grid Index for C*k*NN Queries on Large-Scale Road Networks with Moving Objects

**Kailei Tang, Zhiyan Dong \*, Wenxiang Shi and Zhongxue Gan**

The Academy for Engineering and Technology, Fudan University, Shanghai 200433, China;
20210860147@fudan.edu.cn (K.T.)
\* Correspondence: dongzhiyan@fudan.edu.cn; Tel.: +86-189-4363-1070

**Abstract:** As the Internet of Things devices are deployed on a large scale, location-based services are being increasingly utilized. Among these services, *k*NN (*k*-nearest neighbor) queries based on road network constraints have gained importance. This study focuses on the C*k*NN (continuous *k*-nearest neighbor) queries for non-uniformly distributed moving objects with large-scale dynamic road network constraints, where C*k*NN objects are continuously and periodically queried based on their motion evolution. The present C*k*NN high-concurrency query under the constraints of a super-large road network faces problems, such as high computational cost and low query efficiency. The aim of this study is to ensure high concurrency nearest neighbor query requests while shortening the query response time and reducing global computation costs. To address this issue, we propose the DVTG-Index (Dynamic V-Tree Double-Layer Grid Index), which intelligently adjusts the index granularity by continuously merging and splitting subgraphs as the objects move, thereby filtering unnecessary vertices. Based on DVTG-Index, we further propose the DVTG-C*k*NN algorithm to calculate the initial *k*NN query and utilize the existing results to speed up the C*k*NN query. Finally, extensive experiments on real road networks confirm the superior performance of our proposed method, which has significant practical applications in large-scale dynamic road network constraints with non-uniformly distributed moving objects.

**Keywords:** *k*-nearest neighbor query; moving object; road network





## 1. Introduction

Location-based services have become increasingly important in emergencies, such as COVID-19 or flood disasters, where residents may be confined to their homes or communities [1]. Such services often involve location-based queries, with C*k*NN (continuous k-nearest neighbor) queries being among the most important [2]. The C*k*NN problem in the road network environment is particularly challenging, but socially and commercially valuable [3–5]. For example, the Meituan errand service sends information to several riders closest to the user [6]. In addition, C*k*NN query technology has also promoted the improvement of mobile taxi software, such as Didi and Uber, which can send taxi requests to vehicles close to users [7].

However, several factors make this problem highly challenging. First, the query object dynamically moves within a large-scale road network, which implies that the query response time of urban services is affected by the fast-moving objects [8]. This is further compounded by the fact that objects are continuously on the move [9], thus making the indexing of moment-to-moment moving objects on the road network a major challenge. Secondly, there is a substantial volume of concurrent query requests [10]. For instance, in Beijing, millions of queries are generated every day, and at peak times, tens of thousands of queries per second are processed. Consequently, the need to process high-concurrency queries to identify *k*NN moving objects poses yet another challenge [11]. Furthermore, the existing research scarcely delves into the impact of the distribution of moving objects

in the road network on the performance of neighbor queries. In practical scenarios, the distribution of moving objects in urban environments tends to follow a Zipf distribution; yet, most of the existing studies have not effectively optimized this situation [12].

To address these challenges, this paper presents a novel index, the DVTG-Index (Dynamic V-Tree Double-Layer Grid Index), and a new algorithm, the DVTG-C$k$NN. The DVTG-Index introduces two new vertex types, supports dynamic updates of moving objects, facilitates the allocation of computing resources, and facilitates efficient $k$NN queries. The index initially partitions the entire graph recursively into several grid subgraphs based on the quadtree structure, and subsequently constructs a new grid index to manage these subgraphs. Active vertices associate moving objects with their corresponding network vertices on the road network, and boundary vertices enable the querying of the distance from a point to other vertices in the grid, enhancing computational efficiency.

Our contributions are as follows:

(1) We introduce the DVTG-Index, a scalable, efficient, and adaptable index that accommodates road network datasets and allows for adaptive grid merging and splitting for different regions to obtain the appropriate index granularity and low maintenance cost.

(2) We propose a robust update strategy that supports the dynamic update of moving objects by associating them with vertices to achieve real-time updates of moving objects. This approach better supports the handling of large mobile objects on large-scale road networks for mobile updates.

(3) Building on the DVTG-Index, we design a novel $k$NN search method that leverages the index's capabilities to compute $k$-nearest objects.

(4) We conduct extensive experiments on four real datasets to evaluate our approach's efficacy. Our experimental results demonstrate that our approach significantly outperforms existing baseline methods.

This paper is structured as follows: Section 1 presents an overview of the problem, while Section 2 reviews the relevant literature. Section 3 introduces the method overview and relevant definitions in this work. Section 4 details the DVTG-Index's structure and Section 5 outlines an efficient $k$NN search algorithm. Section 6 presents our experimental results. Section 7 concludes the paper.

## 2. Related Work

The C$k$NN query has been an extensively studied research topic, with numerous theories and algorithms developed to address it. The research at present aims to reduce the query-processing time on the server side and deliver rapid results to users. However, because the continuous $k$NN query involves numerous moving objects, object-by-object scanning considerably reduces the query-processing performance [13]. To overcome this challenge and improve query-processing efficiency, location indexing techniques have been developed and are deemed necessary.

Traditional database indexing sorts the data to store and access records effectively. However, these methods are not appropriate for frequently changing the data or location of moving objects due to the high cost of updates [14]. Moving object location indexing is a challenging research field, with proposed index structures based on spatial, time, or combined spatial and time indexing technologies. The research at present is still in the preliminary stage, both domestically and abroad [15].

The spatial index can be divided into categories according to the basic unit of index structure. Representative indexes include grid, tree-like, and hybrid indexes [16]. In [17], Yang et al. proposed an efficient distributed solution for $k$NN queries, which can handle larger amounts of moving-object data. The proposed solution included a new network-based index called BGI (block grid index). The BGI is an in-memory index based on a distributed hierarchical grid with a minimum and maximum predefined number of moving objects per block. Additionally, Park et al. [18] proposed a DGI (distributed grid index), which builds an index in the form of a hierarchical grid according to the location of objects. S-GRID (scalable grid) is also one of the very classic grid indexes that was proposed by Huang et al. [19] to deal

with the problem of road network indexing. Although these grid-based indexes pre-define the number of moving objects or layer the grid, as the moving objects and road networks change, the index cannot rapidly adapt or produce efficient query responses. This limits the efficiency of these indexes in large, dynamic, road network graphs.

In addition, the most common tree-like indexes are R-Tree, K-D Tree, Z-Tree, and Quad-Tree indexes. Among them, R-Tree and PMR Quad-Tree [20] indexes have a good performance for indexing traditional spatial objects. In order to further improve the query efficiency of $k$NN, Zhong et al. proposed a balanced search-tree G-Tree (grid tree) index structure in 2013 [21]. Based on G-Tree, the G*-Tree index improves the query efficiency of G-Tree by dividing each node into two disjoint regions [22]. This optimization is based on a simple observation that, during querying, when a region is completely contained within the query range, other regions that do not intersect with this region can be ignored. Recently, Bareche et al. [23] proposed the Velocity SpatioTemporal indexing approach. The proposed structure is based on a selective velocity-partitioning method that reduces the update cost and improves the response time and query precision. Bilong Shen [24] of Tsinghua University proposed a new index: V-Tree. It can support a $k$-nearest neighbor search based on the road network. Second, it can support the dynamic updating of moving objects. The V-Tree index's structure is optimized further, based on G-Tree, to become the $k$NN query for moving objects. While these methods have shown a good performance in indexing traditional spatial objects, they may face several limitations when applied to the query of moving objects in large road networks. One limitation is that these methods may not efficiently handle the continuous updates of the location's information of moving objects, resulting in high update costs and query latencies. Another limitation is that they may not consider the road network's topology, which may result in inaccurate $k$NN queries and longer query times. Furthermore, these methods may not take into account the different speeds and directions of moving objects, leading to inefficient queries and inaccurate results. Correspondingly, the index proposed in this paper has an efficient update strategy and can cope with changes in large road networks with low maintenance costs.

There are two problems in the existing study. On the one hand is the question of applicability. Most of the existing methods are based on Euclidean distance constraints [25], and it is difficult to solve the problem of the continuous $k$-nearest neighbor query of moving objects based on road network constraints [26]; however, the distance between moving objects in real life is often determined by the length of the road network. Secondly, some methods are based on the fact that the query point is fixed [27], and only the position of the moving object changes, and then the continuous $k$-nearest neighbor incremental query algorithm of the moving object is designed based on this assumption. Once the position of the query point changes, the query result needs to be recalculated. Another aspect is the issue of efficiency. Although some methods can support the $k$-nearest neighbor query of moving objects based on the road network, when the road network scale is large, the number of query calculations significantly increases, resulting in reduced query efficiency, the poor real-time performance of query results, and insufficient practicability [28]. At times, the structure of the moving object index is complex, resulting in a large-scale update of the location of the moving object [29]; the update efficiency of the moving object's index structure is low, resulting in a decrease in the accuracy of the query result.

## 3. Methodology Overview and Problem Definition

In this section, we introduce the system overview and some relevant definitions for this study.

### 3.1. Methodology Overview

Unlike some existing methods that use tree-like indexes and create rules to optimize the search efficiency, we identified some issues with this approach. First, numerous leaf nodes cannot be efficiently pruned, which leads to numerous unnecessary vertices in these nodes.

Second, not all leaf nodes in the index are used for location-based services, and some of them have a minor effect, but still require significant costs for the construction and updates.

Based on the abovementioned problems, our scheme adopted the overall idea of parallel and balanced computing to build high-quality indexes. As shown in Figure 1, we first recursively partitioned the whole graph into multiple-grid subgraphs according to the quadtree structure, and then constructed new grid indices to take over these subgraphs. The moving object can determine the grid unit to which it belongs according to its position and compile it into the index of the corresponding grid unit. The granularity of meshing is an important factor that directly affects the performance of index building and updating behaviors. If the granularity is too large, there are too many nodes in the grid, and the function of dividing the units is lost; if the granularity is too small, the index needs to be frequently updated, which affects its overall performance.

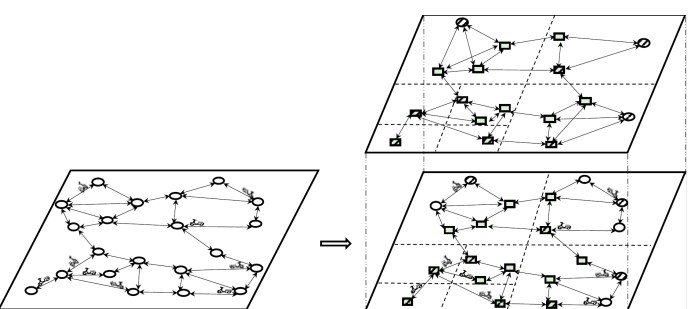

**Figure 1.** Methodology Overview.

*3.2. Problem Definition*

**Definition 1.** *Road Network. In this paper, we abstractly modeled the real road network as a directed weight graph, $G = \langle V, E \rangle$, each road was regarded as an edge of G, and the road endpoints were regarded as the corresponding vertices of G, where V is the set of nodes on the road and E is the set of edges on the road. A vertex on the road is $v_i = (x_i, y_i) \in V$, x is the abscissa of the vertex, y is the ordinate of the vertex; an edge on the road is $e = (u, v, w) \in E$, u is the road. The starting point of e is the end point of the road, and w is the length of the road. The road network used to illustrate the concepts in this paper is shown in Figure 2.*

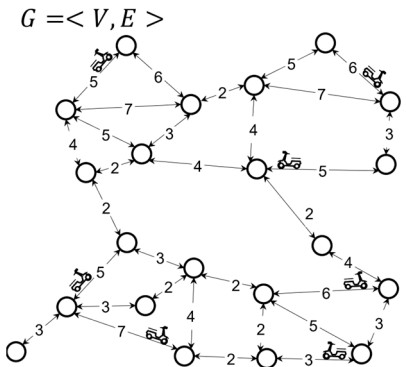

**Figure 2.** Diagram of road network.

**Definition 2.** *kNN Query. The NN query problem originated from the post office problem proposed by Knuth in 1973. The problem can be simply described as: given that set P in the NN -dimensional space contains n data points, find a data point p in set P such that the distance between p and the query point q is the closest. The $kNN(k \geq 1)$ query is a generalized form of the NN query. In this paper, the k-nearest neighbor query is mainly performed on the moving objects in the road network environment, that is, on the road network $G = \langle V, E \rangle$, there is a group of moving objects*

$O_r (|O_r| > k)$; *given a query point q and a positive integer k, find the k moving objects $M(q)$ closest to the q -way network. In addition:*

- $|M(q)| = k$;
- $Mq \subseteq O_r$;
- $\forall o \in O_r,\ o' \in O / M(q),\ Dist(q, o) \le Dist(q, o')$

**Definition 3.** *CkNN Query. In our semantics, moving objects move along the edges of the road network G and periodically report their positions with time interval $\Delta t$. The snapshots $\Delta t$ of all objects are then updated at each time interval. Assuming that the latest snapshot of an object was created at time point $t_s$ and all queries are made at time period $(t_s, t_s + \Delta t]$, the evaluation is first based on this snapshot. Subsequently, the kNN of each query is periodically updated based on each new snapshot of the object. We call queries following this semantics CkNN queries.*

## 4. Dynamic V-Tree Double-Layer Grid Index

In this section, we first define the DVTG-Index and discuss how to construct it. Then, we describe the maintenance strategy, and conclude this section by discussing how to update the DVTG-Index.

### 4.1. Building the Index

To build the index, first we divided graph *G* corresponding to the global road network into d layers according to the quadtree structure, each layer was marked as $L_i$ ($i \in [1, d]$), and the $L_1$ layer was the top layer. The area was divided into 4 grids of the same size, and each grid was divided into four grids at the $L_2$ layer. According to this method, recursively divide until the moving objects contained in the smallest grid do not exceed $\lambda$, and the $L_d$ layer divides the global road network into $4^d$ grids. A grid unit of layer $L_i$ is recorded as $g_i^j$ ($j \in [1, 4^j]$). As shown in Figure 3, each grid $g_d^j$ of the $L_d$ layer covers all the edges and vertices of graph *G* in this region. The divided subgraph is divided into a $4^d$ branch tree, that is, the root corresponds to the whole graph and the leaf matches the smallest subgraph, that is, the underlying grid subgraph.

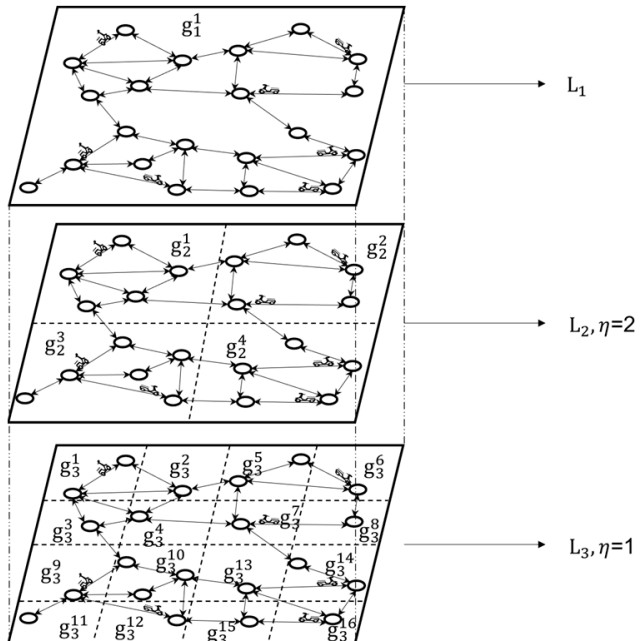

**Figure 3.** Diagram of road network division.

As shown in Figure 3, the second-layer grid index has 4 grid subgraphs, namely, $g_2^1, g_2^2, g_2^3,$ and $g_2^4$, and the third-layer grid index has 16 grid subgraphs.

Then, we constructed the DVTG-Index, which replaces the structure of this subgraph and removes all the subgraphs covered by non-leaf nodes. The DVTG-Index only retains the complete topology of the original graph. Non-leaf nodes are considered virtual nodes and await materialization as they are required to provide indexing services. On the grid subgraph of each leaf node, the shortest distance between several vertices was calculated in parallel, offline, and used as the basic index material for indexing. Furthermore, we introduced two special types of vertices and some related concept definitions for each subgraph.

**Definition 4.** *Active Vertex. If the moving object $O_r$ is on the vertex $v_a$ or the edge $e_{b,a}$ and moves from vertex $v_b$ to vertex $v_a$, then vertex $v_a$ is defined as the active vertex and $O_r$ is associated with $v_a$. Moving objects: active vertex $v_a$ saves and maintains a collection of all moving objects associated with it, denoted as $Q_a$. The vertex set consisting of active vertices in grid $g_i^j$ is denoted as $\mathcal{A}_i^j$.*

**Definition 5.** *Boundary Vertex. If two vertices $v_a$ and $v_b$ on an edge $e_{a,b}$ are given, they do not belong to any grid unit of the same layer $L_i$ at the same time; then, $v_a$ and $v_b$ are both boundary vertices of layer i. The vertex set consisting of boundary vertices in grid $g_i^j$ is denoted as $\mathcal{B}_i^j$.*

**Definition 6.** *$Path(v_u, v_w)$. In a connected path network, there are one or more paths from one vertex $v_u$ to another vertex $v_w$, and the set of all paths is recorded as $Path(v_u, v_w)$. The elements in the collection are the edges in the path, which can be expressed as:*

$$Path(v_u, v_w) = \left\{ (e_{u,a}, \ldots, e_{b,w}), \ldots, (e_{u,x}, \ldots, e_{y,w}) \right\}. \tag{1}$$

**Definition 7.** *$Dist(v_u, v_w)$. In a connected road network, there are one or more paths from one vertex $v_u$ to another vertex $v_w$, and the sum of the distances of all paths is recorded as $Dist(v_u, v_w)$. The elements in the set are the sum of the distances of the edges in the path, which can be expressed as:*

$$Dist(v_u, v_w) = \left\{ (w_{u,a} + \cdots + w_{b,w}), \ldots, (w_{u,x} + \cdots + w_{y,w}) \right\}. \tag{2}$$

**Definition 8.** *$SPath(v_u, v_w), SDist(v_u, v_w)$. In the path set $Path(v_u, v_w)$ between two vertices, $v_u$ and $v_w$, the path with the shortest side length and distance is called the shortest path, denoted as $SPath(v_u, v_w)$, and the distance of this path is denoted as $SDist(v_u, v_w)$.*

Since any path from vertex $v_u$ to vertex $v_w$ outside the grid unit to which it belongs must pass through a certain boundary vertex $x$ of the grid unit, $Dist(v_u, v_w)$ is expressed; therefore, the shortest path of $v_u$ to $v_w$ can be expressed as:

$$SPath(v_u, v_w) = SPath(v_u, x) + SPath(x, v_w). \tag{3}$$

The shortest path distance can be expressed as:

$$SDist(v_u, \ v_w) = SDist(v_u, x) + SDist(x, v_w). \tag{4}$$

When there is an active vertex $v_a$ in the grid, that is, there is a moving object $p_r$ moving towards $v_a$, then $p_r$ is in $Q_a$. Then, the shortest distance between another vertex $v_b$ belonging to the same grid and moving object $p_r$ indexed by $v_a$ can be expressed as:

$$SDist(v_b, \ p_r) = SDist(v_b, v_a) + \delta(p_r, v_a \ ). \tag{5}$$

In the underlying grid $g_d^k$, only the boundary and active vertices in the $k$th grid of its $d$ level are included. Taking the road network in Figure 3 as an example, when $\lambda = 2$, the grid is divided into the second layer, and the division of the $L_2$ layer graph and the settings of active and boundary vertices are shown in Figure 4.

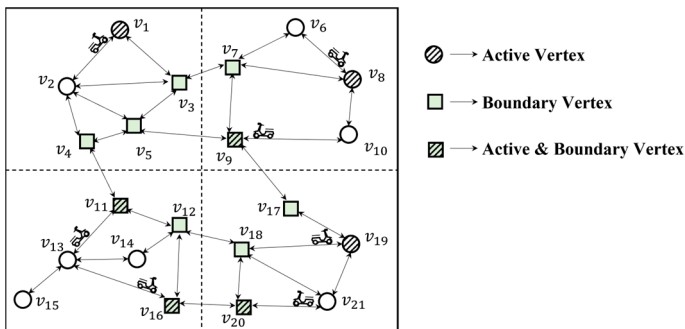

**Figure 4.** Schematic diagram of active and boundary vertices.

As shown in Figure 4, $v_1, v_8$, and $v_{19}$ are active vertices, $v_3, v_4, v_5, v_7, v_{12}, v_{17}$, and $v_{18}$ are boundary vertices, and $v_9, v_{11}, v_{16}$, and $v_{20}$ are both active and boundary vertices.

The DVTG-Index retains the grid subgraph of the leaf nodes; therefore, the grid $g$ can be used to form a graph $G$, and each grid subgraph can be expressed as $g = \langle V_g, E_g \rangle$. The grid subgraphs in the DVTG-Index do not overlap each other, and all grid subgraphs contain all vertices belonging to the graph. Given the two vertices, $v_a$ and $v_b$, belonging to the same grid, we used $RelSDist(v_a, v_b)$ to represent the shortest distance between $v_a$ and $v_b$ within the grid, which was the relative shortest distance. Relatively, $SDist(v_a, v_b)$ represents the shortest distance between $v_a$ and $v_b$ relative to the whole graph, that is, the absolute shortest distance. It should be noted that the shortest distance expressed by $RelSDist$ is only relative to the grid subgraph, not necessarily the shortest distance in the whole graph. Therefore, $RelSDist(v_a, v_b)$ is not necessarily equal to $SDist(v_a, v_b)$; it should satisfy:

$$RelSDist(v_a, v_b) \geq SDist(v_a, v_b). \tag{6}$$

To make searching for grids more efficient, we indexed the shortest distance between the boundary and active vertexes in each grid. This allowed us to access the internal active vertex and its associated objects through any boundary vertex, which reduced the detection cost of the grid. We also used active nodes and boundary vertices to help identify the $k$NN. To further optimize the index structure, we introduced the concepts of vertex subgraph, vertex distance matrix, and vertex road network graph. The vertex subgraph only included active vertices, boundary vertices, and the shortest distance to a leaf node.

**Definition 9.** *Vertex Subgraph. Let $v_{i,a}^j$ and $v_{i,b}^j$ be active and boundary vertices, and the active and boundary vertices sets in the grid $g_i^j$ are marked as $\mathcal{A}_i^j$ and $\mathcal{B}_i^j$; the vertex subgraph is recorded as $ChiG_i^j = \langle V_i^j, E_i^j, w_i^j \rangle$, which satisfies:*

- $V_i^j = \mathcal{A}_i^j \cup \mathcal{B}_i^j$;
- $E_i^j$ *represents the edge between $V_i^j$, then $e_{i,x,y}^j$ is $v_{i,a}^j$ and $v_{i,b}^j$ and other boundary vertices, that is, $e_{i,x,y}^j = SPath\left(v_{i,x}^j, v_{i,b}^j\right), v_{i,x}^j \in V_i^j$;*
- $w_i^j$ *represents the distance of side $E_i^j$, that is, $w_i^j = SDist\left(v_{i,a}^j, v_{i,b}^j\right) = min\left\{(w_{a,m} + \cdots + w_{n,b}), \ldots, (w_{a,u} + \cdots + w_{w,b})\right\}$.*

As shown in Figure 5, $v_2$ is neither an active nor a boundary vertex. When searching inside the road network, we can prune such nodes; therefore, only active and boundary vertices are kept when constructing a vertex subgraph, while the edge in the graph is a virtual edge that represents the shortest route between two points and does not represent the actual meaning. The weight value of the edge is the distance of the shortest route.

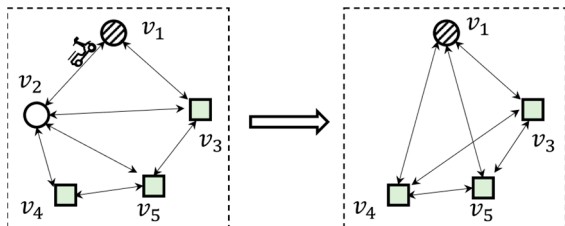

**Figure 5.** Vertex subgraph diagram.

**Definition 10.** *Vertex Distance Matrix. Each vertex subgraph $ChiG_i^j$ needs to maintain a vertex distance matrix, saving the shortest distance $SDist(v_a, v_b)$ between each active vertex and each boundary vertex. This distance matrix is recorded as $DistM_i^j$, which satisfies:*

$$DistM_i^j = \begin{bmatrix} SDist\left(v_{i,a_1}^j, v_{i,b_1}^j\right) & SDist\left(v_{i,a_1}^j, v_{i,b_2}^j\right) & \cdots & SDist\left(v_{i,a_1}^j, v_{i,b_m}^j\right) \\ SDist\left(v_{i,a_2}^j, v_{i,b_1}^j\right) & SDist\left(v_{i,a_2}^j, v_{i,b_2}^j\right) & & \\ & \vdots & \ddots & \vdots \\ SDist\left(v_{i,a_{n-1}}^j, v_{i,b_1}^j\right) & SDist\left(v_{i,a_{n-1}}^j, v_{i,b_2}^j\right) & \cdots & SDist\left(v_{i,a_n}^j, v_{i,b_m}^j\right) \\ SDist\left(v_{i,a_n}^j, v_{i,b_1}^j\right) & SDist\left(v_{i,a_n}^j, v_{i,b_2}^j\right) & & \end{bmatrix}. \quad (7)$$

**Definition 11.** *Vertex Elevated Graph. All sub-grids at the bottom layer can form a graph G. Similarly, all vertex subgraphs form a vertex elevated graph. Let $ChiG_i^j$ be the vertex subgraph, and the vertex elevated graph is recorded as $G_v = \left\{ ChiG_d^1, ChiG_d^2, \ldots, ChiG_d^{4^d} \right\}$.*

As the construction of the vertex subgraph is completed, at this time, based on all the vertex subgraphs in the graph, the vertex elevated graph $G_V$ is constructed, and $G_V$ includes all active and boundary vertices in the vertex subgraph.

The constructed vertex elevated graph considerably reduces the number of vertices and edges, thus providing efficient services for subsequent neighbor queries. Figure 6 presents the pruning process for constructing a vertex elevated graph. Taking the road network graph in Figure 4 as an example, we set the threshold $\eta$ to 2, divided the road network graph into 4 grid subgraphs, reconstructed the active and boundary vertices in each grid, and then performed pruning. As shown in Figure 4, the original road network graph has 21 vertices and 32 edges, and the Vertex Elevated Graph after the pruning strategy has only 14 vertices and 24 edges.

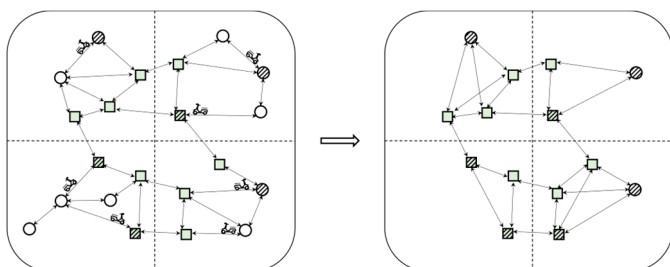

**Figure 6.** The comparison between the vertex subgraph and original road network graph.

We built vertex subgraphs for all underlying grids, which were then continuously maintained for the real-time reporting of the motion of moving objects, updating active vertices, boundary vertices, and the vertex distance matrix $DistM_i^j$. However, in fact, we found that many vertex subgraphs may not be useful for a certain query; however, updating it incurs a certain maintenance cost. Therefore, after constructing the vertex subgraph for the underlying grid subgraph, we will continue to scale the relevant vertex subgraph

according to the movement and distribution of the moving objects to adjust the index granularities of different regions. The specific maintenance and update mechanisms are discussed in detail in the subsequent section.

### 4.2. Update Mechanism of the DVTG-Index

Since the *k*NN problem we discussed in this study was based on moving objects in the road network environment, we needed to update the DVTG-Index as the moving objects frequently change in the road network. The update was divided into two parts: the first part updates the position of the moving object, that is, it updates the moving object $O_r$; the second part changes the distribution of the moving objects, active vertices in the grid, and the division granularities of moving objects. We need to further divide the grid according to the threshold, so as to build a better vertex subgraph.

### 4.2.1. Update of Moving Objects

The real-time update of large moving objects is a major feature in the urban road network environment, and it is also a difficulty in this research scenario. This paper introduced the concept of the active vertex; therefore, binding the moving object to the corresponding active vertex can efficiently handle the problem of updating the position of the moving object. The specific details of the update strategy are:

First, in the road network, the motion scenes of moving objects can be summarized into the following three situations:

- Add moving object: we added a new object $O_r$ on edge $e_{u,v} = (u, v, w)$, and $O_r$ moved towards vertex $v$, for example, a taxi driver just exited the car or just finished the last task.
- Delete a moving object: we deleted a moving object $O_r$ from the edge $e_{u,v} = (u, v, w)$, for example, the driver finished a task and left work, or the car broke down and could no longer carry passengers.
- Update the moving object: the position of the moving object $O_r$ changed, which was the most common situation. We further subdivided it into two cases according to the variation in moving objects. The first case was that $O_r$ was still on the edge $e_{u,v}$; however, the distance from the active vertex $v$ changed. In this case, we only needed to change the $\delta$ attribute in $O_r = (t, (u, v), \delta)$, and the time complexity was only O(1); the second case was that the side where $O_r$ was located changed, for example, $O_r$ moved from side $(u, v)$ to side $(v, w)$. At this time, we only needed to combine the operations of (1) and (2), delete $O_r$ from edge $u$, and add $O_r$ on edge $(v, w)$, that is, the cross-edge movement of the moving object was realized.

From the analysis above, we only needed to consider the operations of adding and deleting moving objects.

Add moving object: add moving object $O_r$ to edge $e_{u,v} = (u, v, w)$. At present, there are two cases according to the state of the vertex. (1) If $v$ is an active vertex, you only need to add $O_r$ to the moving object collection $Q_v$ maintained by $v$; (2) if $v$ is not an active vertex and $Q_v$ is an empty set, we first mark $v$ as an active vertex, and at the same time add $O_r$ to the moving object collection $Q_v$ maintained by $v$. Then, judge whether the number of moving objects in the vertex subgraph where $v$ is currently located is within the threshold range. If it is within the range, the vertex subgraph does not need to be split or merged. Then, calculate the shortest distance from this point to other boundary vertices in the vertex subgraph and update the shortest distance matrix. If not within the threshold range, iteratively split or merge vertex subgraphs, and then calculate the shortest distance matrix of the new vertex subgraph.

Remove moving objects: remove $O_r$ from edge $e_{u,v} = (u, v, w)$. At this point, two situations still need to be considered: (1) if the number of moving objects contained in the moving object set $Q_v$ maintained by vertex $v$ is more than 1, then it is only needed to delete $O_r$ from $Q_v$; (2) if there is only one moving object $O_r$ in $Q_v$, then we also delete $O_r$ from $Q_v$ first, because $Q_v$ is empty at this time, that is, no moving object moves to $v$,

then $v$ becomes a normal vertex. Then, it is also judged whether the number of moving objects in the vertex subgraph where $v$ is currently located is within the threshold range, and corresponding merging and splitting operations are performed.

### 4.2.2. Index Update

Updates to moving objects cause the number of active vertices within the vertex subgraph to change. At present, the distribution of moving objects in the city is mostly a Zipf distribution, not an even distribution; therefore, the number of active vertices in different vertex subgraphs is quite different, and the corresponding vertex distance matrix also requires higher calculation and space costs. In order to more reasonably balance the global computing load, the vertex subgraph needs to merge or split the subgraph according to the relationship between the number of active vertices of itself and the surrounding and set thresholds. In this study, the threshold value of number of active vertices $v_a$ in the subgraph was set to $\eta$.

When adding a moving object, if the number of active vertices in the current vertex subgraph $g_j^i$ is more than $\eta$, the subgraph g is split into four subgraphs, and furthermore, the four subgraphs of g are updated to calculate the corresponding shortest distance matrix. If the number of moving objects in a certain subgraph still meets the split condition, continue to recurse the abovementioned operations, layer by layer, until the split condition is not met.

When deleting a moving object, if the sum of the number of active vertices in the current vertex subgraph $g_j^i$ and the number of active vertices in its three sibling vertex subgraphs is less than the specified threshold $\eta$, merge them into a new vertex subgraph $g_{j'}^{i-1}$. If the number of active vertices of the parent subgraph $g_{j'}^{i-1}$ of $g_j^i$ and its three sibling subgraphs is still less than $\eta$, continue to recursively operate layer by layer until the number of active vertices in the upper subgraph $g_{j'}^i$ of $g_j^d$ $(1 \leq i < d-1)$ is not less than $\eta$ or $i = 1$. $g_{j'}^{i-1}$. Save and maintain all the active vertices in the four subgraphs, update the border vertices, and recalculate the shortest path $SPath(v_u, v_w)$ between any border and other vertices; save the shortest distance $SDist(v_u, v_w)$ between vertices and the vertex distance matrix $DistM_i^j$.

When the road network map is large enough, there will be a high number of moving objects that need to be updated at the same time. Therefore, we used the method of scanning subgraphs to update all the moving objects in batches, and then updated the vertex subgraphs, which greatly improved the update efficiency of the DVTG-Index.

According to the description of the index update algorithm, if updating the moving object does not change the attributes of the vertices, the time complexity of updating is $O(1)$. If the update operation changes the attributes of each vertex and changes the index structure, it needs to be discussed separately. Since the vertex subgraph needs to be recursively split or merged every time a moving object is added/deleted to balance the computational load, the time complexity of each adding/deleting a moving object is related to the recursive depth of the vertex subgraph. Therefore, it can be observed that the time complexity of the algorithm is $O(\log n)$. Because each operation may cause the recursion depth of the vertex subgraph to change, the time complexity can only be used as an approximation.

The index update algorithm can effectively balance the global computing load; however, it also needs to consider the impact of space complexity, because splitting subgraphs increases the space usage. We tested this in the experiments in the subsequent section.

With the update of the index of the change in the position of the moving object, taking the road network in Figure 7 as an example, when there are more moving objects in the grid subgraph in the lower left corner of the original road network diagram, as shown in Figure 5, the number of active vertices reaches three, which exceeds the threshold. According to the index update mechanism, the grid continues to be divided, and then corresponding pruning is performed to form a new vertex subgraph. The updated vertex elevated graph

has 17 vertices and 28 edges. Compared with the original road network graph, the number of the overall vertices and edges is still effectively reduced, and the number of active vertices in a single vertex subgraph is also more balanced, which reasonably allocates the computing cost in the subgraph and improves the overall data-processing capability.

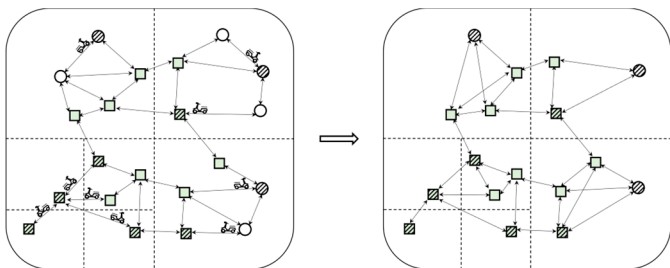

**Figure 7.** The schematic diagram of index update instructions.

## 5. DVTG-C*k*NN Query Algorithm

We discussed the basic concept of the *k*NN query in the second section, and this section describes the process of the *k*NN query algorithm in detail. The *k*NN query problem that is usually studied is a single query, that is, given the position of the query point q, from graph G, find the *k* moving objects closest to q. However, in daily life, there are often multiple and continuous queries. In the same location, especially in the downtown area, there are often multiple query requests. If we regard each query request as a single query, it produce numerous repeated calculations, which will affect the efficiency of the *k*NN query. Therefore, we proposed the DVTG-C*k*NN algorithm based on the DVTG-Index. For each first query request, we used the DVTG-*k*NN algorithm to calculate the initial *k*NN of query q. Then, with the movement of the moving object, the *k*NN of the continuous query request q was updated round by round in the incremental search mode, and the search range of the initial query was saved, so that when the position was queried again, effective incremental calculations could be performed to improve the efficiency of the algorithm.

### 5.1. Query-Point Processing

Due to the randomness of the query points, we needed to preprocess the query points. The processing method is similar to the method used for processing moving objects in Section 4.2.1. We associated the location of query request q with the vertices in the road network.

In summary, the query point *q* query moving object problem in the road network was transformed into a vertex query problem in the road network.

### 5.2. The First k-Nearest Neighbor Query

According to the above points, we associated the query point with the vertices of the vertex subgraph, and the query between objects was converted into the query between vertices. In order to better explain the query algorithm, we first need to define some concepts.

**Definition 12.** *Vertex Queue S. We defined vertex queue S as:*

$$S = \{q\} \cup \left\{ v \in V \left| \begin{array}{c} v \text{ is a boundary or active vertex} \\ \text{in the vertex subgraph where } q \text{ resides} \end{array} \right. \right\}. \tag{8}$$

During the search process, the vertices in the processed vertex subgraph are continuously added to *S* until the algorithm terminates or the graph *G* is traversed. We can express this process as follows:

$$S_{t+1} = S_t \cup \{v \in V | v \text{ is a vertex in the processed subgraph at iteration } t\}. \tag{9}$$

where $S_{t+1}$ denotes the state of the vertex queue at iteration *t*.

Since graph $G$ is a directed and weighted graph, all road networks can be connected through boundary vertices. The $S$ queue represents the scope of the search map at present. During the search process, the vertices in the processed vertex subgraph are continuously added to $S$ until the algorithm terminates or the graph $G$ is traversed.

**Definition 13.** *Query Queue $F_q$. The query queue $F_q$ stores the n moving objects closest to the query point q, that is, the final result of the k-nearest neighbor query. In $F_q$, the road network distance between the moving object $O_n$ that is farthest from q and q is denoted by $\zeta$. Therefore, we can mathematically represent $F_q$ as follows:*

$$F_q = \{O_i | i = 1, 2, \ldots, n\}. \tag{10}$$

*where $O_i$ is the i-th moving object in the queue $F_q$. The distance between each moving object $O_i$ and query point q is denoted by $SDist(O_i, q)$. Let $\zeta$ be the maximum distance between any moving object $O_i$ and q. Therefore, we can express $\zeta$ as:*

$$\zeta = \max(SDist(O_i, q)). \tag{11}$$

When the algorithm ends, the distances from other mobile objects $O_n$ to the query point $q$ are greater than $\zeta$, that is:

$$SDist\big(O_n{}', q\big) \geq \zeta, \forall\, O_n{}' \notin F_q. \tag{12}$$

**Definition 14.** *Query Vertex Queue D. The query vertex queue D stores the vertices that have been determined to have the shortest distance to the query point. Queue D is initially created as an empty set. The vertices are added to D during the search process as their shortest distances are determined. Mathematically, we can define D as:*

$$D = \{v | v \text{ is a vertex that has been determined to have the shortest distance to } q\}. \tag{13}$$

Queue $D$ is convenient for subsequent continuous queries, because the boundary vertices are not updated as the position of the moving object changes. Therefore, the border vertex data in the queue can still be used again in continuous queries, and the number of border vertices is higher than that of active vertices; therefore, the repeated traversal of vertices can be reduced.

Then, we detailed the steps of the first *k*-nearest neighbor query algorithm:

When the system received a *k*-nearest neighbor query request $(q, k)$, it first associated query point q with its nearest vertex v; if $v$ is not an active or boundary vertex of the vertex subgraph, it means that vertex $v$ is not in the vertex set of the vertex subgraph, and the distance from vertex $v$ to other vertices in the vertex subgraph is not calculated. Therefore, we needed to activate $v$ as an active vertex first, add it to the vertex set of the vertex subgraph, update the index, and obtain the shortest distance between $v$ and any boundary vertex in the vertex subgraph and the active vertex. If $v$ itself is an active or boundary vertex in a vertex subgraph, no additional operations are required. The boundary and active vertices in this vertex subgraph are then added to vertex sequence $S$.

Then, initialize query queue $F_q$ to be empty, and traverse all vertices in $S$, find the vertex $v_1$ closest to vertex v, and start processing vertex $v_1$ (the first vertex to be processed is the query vertex $v$ itself, because the weight of the road is a positive number and $SDist(v, v) = 0$; therefore, there is no need to consider the vertex itself). If $v_1$ is the active vertex, add $v_1$'s moving object set $Q_v$ to $F_q$, and the distance from the moving object $O_n$ in the set to the query point q is $SDist(v_1, v) + \delta$; if $v_1$ is a border vertex, it is necessary to expand the search area and traverse all adjacent border vertices u of $v_1$. If u is in vertex queue S and $SDist(u, v_1) + SDist(v_1, v) < SDist(u, v)$, update $SDist(u, v) = SDist(u, v_1) + SDist(v_1, v)$. If $u$ is not in vertex queue S or query vertex queue D, add $u$ to S. At this point, the processing of vertex $v_1$ is completed, and then $v_1$ is deleted from $S$ and added to query vertex queue $D$. Then, traverse all the nodes in $S$ again, and continue to process the subsequent vertex closest to

query vertex $v$ until the algorithm ends early or $S$ is empty (if the algorithm ends early, it means that there are $k$ or more moving objects in the query queue, and if $S$ is empty, it means that all the connected graphs have been searched).

If the number of moving objects in $F_q$ is greater than $k$, more than $k$ moving objects have been found (because the number of moving objects contained in each active vertex is different, the number of moving objects saved in $F_q$ will not be exactly $k$ every time), then sort them according to the road network distance from the query point. Whenever the subsequent vertex $v_n$ is processed, a judgment is made: if $SDist(v, v_n) \geq \zeta$, the algorithm ends early. Because the Dijkstra-style algorithm searches for vertices in the order of near to far, if $SDist(v, v_n) > \zeta$, the distance from the subsequent active vertex $u$ containing the object moving to the query vertex $SDist(v, u)$ must be greater than or equal to $SDist(v, v_n)$; therefore, the distance between the object $O_n$ moving towards $u$ and the query point must be greater than or equal to $SDist(v, u)$, and thus greater than $\zeta$. From this, it can be determined that $k$ moving objects in $F_q$ are the $k$ moving objects closest to $q$. If $SDist(v, v_n) < \zeta$ and the processed node is an active vertex, the moving object $O_n$ in this point may still be closer to the query point than the moving object in $F_q$; then, we need to compare $O_n$ with the moving object in $F_q$ and update it. The pseudocode of the first $k$-nearest neighbor algorithm is shown in Algorithm 1.

---

**Algorithm 1.** The first $k$-nearest neighbor query

---

Input: $q,k$ // Find the $k$ closest moving objects to the query point $q$
Output: $F_q$ // Returns the $k$-nearest neighbor query queue

---

*function* $kNN(q, k)$
*Associate q with vertex v nearby*
*add v into $G_V$;*
*add v into S;*
$F_q \leftarrow \varnothing$, $\zeta \leftarrow \infty$;
$D \leftarrow \varnothing$;
*while* $S \neq \varnothing$ *do*
    $Dist \leftarrow \infty$;
    *for* each vertex u in S *do*
        *if* $SDist(v, u) < Dist$ *then*
            $Dist \leftarrow SDist(u, v)$;
    *if* $|F_q| = k$ *and*
        $\zeta \leq Dist$ *then*
        *break*;
    *if* u is in $\mathcal{A}$ *then*
        *add* $u.Q_u$ into $F_q$
        $\zeta \leftarrow min\{\zeta, max\{SDist(O_n, v)\}\}$
    *else*
        *for* each vertex w in u's neighbors *do*
            *if* $w \in S$ and $SDist(v, w) > SDist(v, u) + SDist(u, w)$ *then*
                $SDist(v, w) = SDist(v, u) + SDist(u, w)$
            *else if* $w \notin D$ *then*
             *add w into S*
    *remove u from S*
    *add u into D*
*return* $F_q$

---

## 5.3. Continuous k-Nearest Neighbor Query

Since the boundary vertices are only related to the road network itself, as long as the grid division does not change, the properties of the vertices as boundary vertices will not change. As the position of the moving object changes, only the active vertices are changed. Additionally, the number of boundary vertices is far greater than that of active vertices. Therefore, when query request $q$ has already performed the first $k$-nearest neighbor query

operation, we can simplify the continuous *k*-nearest neighbor query based on the query vertex queue *D*. Perform the query from the range that was searched the previous time, and continue to expand it on the basis of this queue, instead of starting from scratch. The specific steps of the continuous *k*-nearest neighbor query algorithm are as follows.

We first associated query point q with the nearest vertex v, similar to the first *k*-nearest neighbor value. Then, traverse the vertices in query vertex queue *D* saved after the initial *k*-nearest neighbor query operation; if the vertex *v* in it was the active vertex at the time of the last query and is still the active vertex at this time, add the moving objects in the moving object sequence $Q_v$ maintained by *v* to the new $F_q$ . If *v* is no longer the active vertex, proceed to the subsequent node. If *v* is a boundary vertex, traverse the adjacent vertices of *v* to determine whether there is a new active vertex u or a boundary vertex that does not belong to *D*. If so, add it to *S*. If it is an active vertex, judge whether the active vertex is less than the distance from the last vertex in *D* to query vertex *v*, which is the furthest distance from the query point among the confirmed nodes. If it is less than, then update *v*'s mobile object sequence $Q_v$ to $F_q$ , delete u from S, and insert it into *D*.

If the number of moving objects in $F_q$ exceeds *k*, and the distance from the processed vertex in *D* to *v* is greater than the distance $\zeta$ from the moving object farthest from the query point in $F_q$ to the query point, the algorithm is terminated early. If the algorithm still cannot end after traversing all the nodes in *D*, continue to follow the operation of the initial *k*-nearest neighbor algorithm to traverse query range sequence *S* and continue to expand the search range. The pseudocode of the continuous *k*-nearest neighbor query algorithm is shown in Algorithm 2.

---

**Algorithm 2.** Continuous *k*-nearest neighbor query

---

Input: $O_r$ ,*v*  // Find the *k* closest moving objects to the query point *q*
Output: $F_q$ // Returns the *k*-nearest neighbor query queue

---

*function CkNN(q, k)*
*Associate q with vertex v nearby*
*add v into $G_V$*
$D \leftarrow v.D$ , $S \leftarrow v.S$
$F_q \leftarrow \varnothing$
**for** *each vertex u in D or S* **do**
   **if** *u $\notin \mathcal{A} \cup \mathcal{B}$ at this time* **then**
      *remove u from D or S*
*Dist $\leftarrow$ SDist(v, u)*
**for** *each vertex u in D* **do**
   **if** *u $\in \mathcal{A}$* **then**
      *add u.$Q_u$ into $F_q$*
      $\zeta \leftarrow min\{\zeta, max\{Dist(p\_r, q)\}\}$
   **else**
      **for** *each vertex w in u's*
*neighbors* **do**
         **if** *w $\in \mathcal{A}$* **then**
            **if** *SDist(u, w) > Dist* **then**
              *add w into S*
            **else**
              *insert w into D*
   **if** $|F_q| = k$ *and $\zeta \leq Dist$* **then**
      **break**
**return** $F_q$

---

### 5.4. Time and Space Complexity of the DVTG-CkNN Algorithm

5.4.1. Time Complexity

Subsequently, let us analyze the time complexity of the DVTG-*k*NN algorithm. The association of a query point with its nearest vertex is a constant-level time complexity.

The act of updating the vertex subgraph and deriving distances can be viewed as an operation for each active vertex with a time complexity of $O(N)$, where $N$ is the number of all vertices indexed at the top level. However, the time complexity of calculating the distance of each vertex to other vertices is $O(M^2)$; however, since the shortest distance between vertices was obtained by distributed computing in our method, $M$ should be the largest number of vertices in all vertex subgraphs.

Then, the time complexity of enqueuing the vertex is $O(1)$. The time complexity of the operation of traversing the vertex queue is $O(N)$, where $N$ is the number of vertices in the vertex queue. The time complexity of finding vertex v is $O(log N)$, and the minimum heap is used for searching.

For the processing of vertex $v$, if it is an active vertex, the time complexity of adding its moving object set to the query queue is $O(1)$; if it is a border vertex, the time complexity of traversing all its adjacent border vertices is $O(K)$, where $K$ is the degree of each border vertex.

Therefore, the total time complexity is $O(M^2 + N \log N + KP)$, where $N$ is the number of all vertices in the top-level index, $M$ is the number of vertices at most in all vertex subgraphs, $K$ is the degree of each border vertex, and $P$ is the number of border vertices. However, in practice, the time complexity of the *k*NN algorithm also depends on the value of $k$. When $k$ is small, the time complexity of the algorithm is lower; when $k$ is large, the time complexity of the algorithm is higher. Therefore, choosing an appropriate value for $k$ has a great influence on the performance of the algorithm, which we tested and proved in subsequent experiments.

However, the time complexity of DVTG-C*k*NN is difficult to accurately evaluate because it depends on the number of active and boundary vertices close to the query point and the relationship between the vertices. Ideally, the number of active vertices is less than the number of boundary vertices, and the number of adjacent vertices of each boundary vertex is also small. In this case, the time complexity of the algorithm can be assumed to be $O(k)$. However, if the number of active vertices is high and the number of border vertices adjacent vertices is also high, the time complexity may be high. Therefore, the exact time complexity of this algorithm cannot be provided, and we tested it in the experiments in the subsequent section.

5.4.2. Space Complexity

Based on the algorithm's description, the space complexity of the DVTG-C*k*NN algorithm can be analyzed as follows:

The space required to store the vertex subgraph, including active, adjacent, and boundary vertices, can be considered as $O(|V|)$, where $|V|$ is the total number of vertices in the subgraph.

The space required to store the moving object sets of active vertices, including the query queue $F_q$, can be considered as $O(|O|)$, where $|O|$ is the total number of moving objects in the subgraph.

The space required to store vertex sequence $S$ and query vertex queue $D$ can also be considered as $O(|V|)$. In addition, the space required to store the shortest distances between vertices, which are updated during the search, can be considered as $O(|V|^2)$.

Therefore, the total space complexity of the DVTG-C*k*NN algorithm can be approximated as $O(|V|^2 + |O|)$.

## 6. Experiment Analysis

In this section, we experimentally evaluated the performance of DVTG-Index and presented the performance of our proposed DVTG-C*k*NN algorithm when dealing with the *k*NN and continuous *k*NN queries. To fully evaluate the performance of our method, we chose three advanced methods, $G^*$-Tree, V-Tree, and SILC, as benchmark algorithms to compare to our method.

### 6.1. Experiment Settings

This experiment was based on the Java language program, and the IDE program used IDEA. The computer software/hardware configuration is shown in Table 1.

**Table 1.** The computer software/hardware configuration.

| Software and Hardware | Version/Model |
| --- | --- |
| Operating system | Windows 10 (1902) |
| CPU | Intel Core i7-7700 HQ |
| Memory | 8 GB |
| Hard disk | 1 TB |
| Java | Jdk1.8.2 |

Additionally, we validated the performance of our query model by simulating a set of objects moving on a real road network. Therefore, we needed three types of datasets that were real road network, moving-object, and query-request datasets. This experiment selected the following three real-world road networks for the experiments: BJ (Beijing), NY (New York), COL (Colorado), and NW (Northwest USA) [30]. These networks have been widely used in previous studies, and the number of nodes in the road network ranges from hundreds of thousands to millions. The specific information is shown in Table 2.

**Table 2.** Experimental software and hardware configuration table.

| Dataset | Name | Number of Vertices | Number of Sides |
| --- | --- | --- | --- |
| BJ | Beijing | 188,229 | 436,648 |
| NY | New York City | 264,346 | 733,846 |
| COL | Colorado | 435,666 | 1,056,066 |
| NW | Northwest USA | 1,207,945 | 2,840,208 |

For the BJ road network set, we used a real-world dataset, BJ T-drive, containing actual taxi trajectories in Beijing from February 2 to August 2008, which included nearly 18 million GPS trajectories for 10,350 taxis. We used existing methods to map the locations of moving objects and queries onto road networks [31].

For the other three road network sets, we used the moving-object simulation generation software invented by Thomas Brinkoff [32] to realize the initial distribution and update the movement and query requests of moving objects in the road network. We temporarily set the total number of moving objects as one percent of the number of vertices, and we use the following three strategies to simulate the starting position and subsequent trajectories of the moving objects: RD (random distribution), ND (normal distribution), and ZD (Zipf distribution).

### 6.2. Baseline Algorithm

We compared the DVTG-Index with three benchmark methods: V-Tree, G*-Tree, and SILC. These three methods were implemented in this experiment; the implementation of the first two methods was provided by their authors, and the implementation of SILC as completed by us. This is how it worked: we started by building an index structure for the (static) road network. Then, we created a reference index, such as the DVTG-Index, which saves a list of moving objects from the vertex to the vertex. Similar to our approach, if a vertex has some moving objects, SILC calls it a POI. Because POIs may contain multiple moving objects, POIs need to build a list of moving objects. Additionally, when looking for $k$NN moving objects, the algorithm will stop. SILC uses a best-first approach to compute $k$NN results. We also used this index to keep POIs, calculate top-$k$NN POIs, and add moving objects of interest points to the result set.

### 6.3. DVTG-Index Experiment Situation

In this section, we conducted experiments by changing the threshold $\eta$ of the number of moving objects in the parameter grid; we conducted performance evaluations from two aspects: the time and space overhead of the index construction and the time and space overhead of the index update. In order to exclude the influence of other factors, the results of this experiment are the average values obtained after repeated experiments.

First, we evaluates the characteristics of the DVTG-Index, built a static DVTG-Index, V-Tree, G*-Tree, and SILC on the datasets BJ, NY, COL, and FLA, and compared the time cost $t_c$ and space cost of building the index $m_c$, as shown in Figure 8. Among them, the road network set BJ randomly selected the real distribution of taxis at 8:00, 12:00, and 17:00 on February 2, and the road network set NY used three different distribution strategies to set the starting position of the moving object.

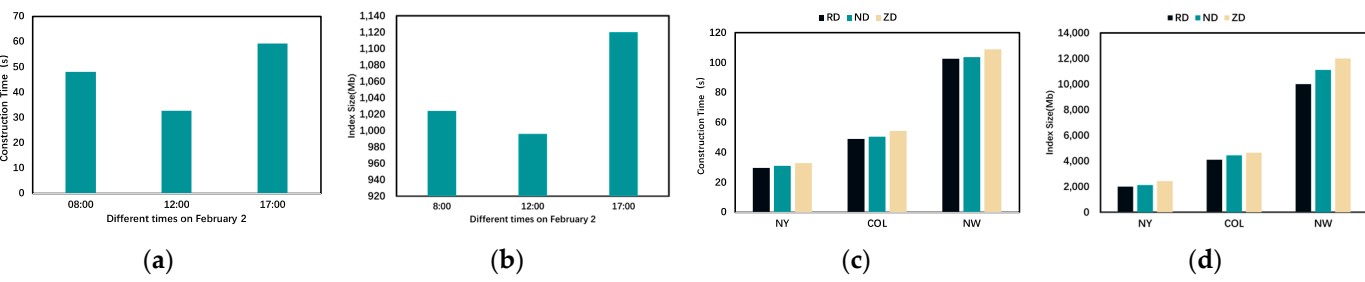

**Figure 8.** Time and space costs of the DVTG-Index. (**a**) Time costs on BJ; (**b**) Space costs on BJ; (**c**) Time costs on NY, COL and NW; (**d**) Space costs on NY, COL and NW.

Among them, the construction time of the DVTG-Index was composed of the network construction time, vertex distance matrix calculation time, and vertex elevated graph construction time; construction space was composed of the road network set, boundary vertices and active vertex sets, vertex distance matrix, and vertex elevated graph. As shown in Figure 8a,b, 8:00 and 17:00 in Beijing are the morning and evening peak periods, respectively; therefore, there are more moving objects and increased time and space factors for the index construction. As shown in Figure 8c,d, the number of vertices and edges of the road network sets NY, COL, and NW increases sequentially. The constructed DVTG-Index needs to construct more vertex subgraphs and calculate more vertex distance matrices at the same time, so that the memory space occupied by the index increases sequentially and the construction overhead time also increases. In addition, different distribution strategies of moving objects also have a certain impact on time and space costs. Under the three distribution strategies of RD, ND, and ZD, the time and space costs of the constructed DVTG-Index increase in turn. With an uneven distribution, the indexed grid requires more merge-up and split-down operations, thus requiring more time and space overhead.

When the index is constructed, the number of grid divisions determines the number of grids in the initial index. According to the above factors, if the grid is divided $l$ times, the number of grids is $4^l$. The number of grids determines the size of the grid area and determines how many nodes and moving objects are included in the grid; therefore, we changed the number of grid divisions $l$ to evaluate the time and space overhead of the DVTG-Index on each dataset. To exclude the influence of other factors, in this experiment, we adopted the ZD distribution strategy, and the data shown in the chart are the average of the results of multiple experiments.

As shown in Figure 9, it can be observed from the experimental results that there is no positive or negative correlation between the construction time and space costs and the number of grid divisions $l$. As the number of grid divisions gradually increases, the time and space values overhead first decrease and then increase. This is because when the number of grid divisions is low, the number of grids is low, the area of a single index grid is large, and the number of active vertices in a single index grid is high, which makes the calculation time and space overhead of the shortest distance matrix in the grid larger. Due

to the small number of index layers, the calculation time and space overhead of the shortest distance matrix account for a large proportion of the total cost; therefore, as the grid is split, the total time and space costs of the index construction are significantly reduced. However, when the number of index layers gradually increases to a certain number, the time to split the index grid down gradually increases, and the number of shortest-distance matrices increases, which also makes its space cost increase sharply; therefore, the total time and space overhead of the index construction starts to increase as the number of index layers increases. In addition, we found that the experimental results of the time and space overhead of building indexes have similar trends in different datasets; therefore, we can use the NY dataset to evaluate the index performances in subsequent experiments.

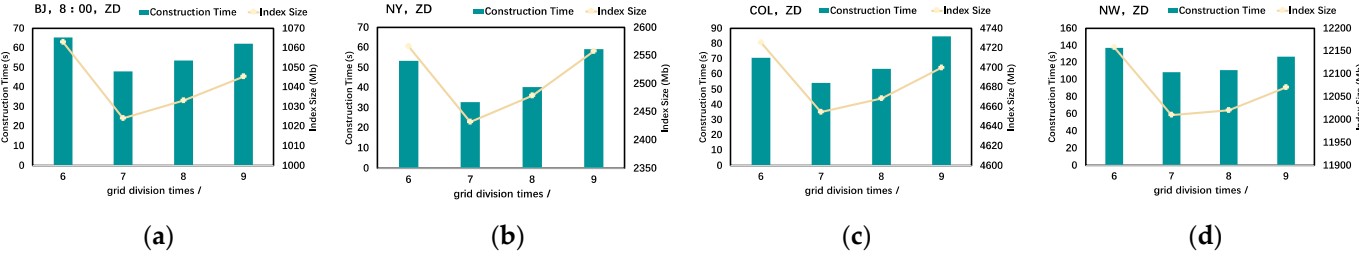

**Figure 9.** Construction time and space costs overhead of DVTG-Index w.r.t. *l.* (**a**) Construction time and space costs on BJ; (**b**) Construction time and space costs on NY; (**c**) Construction time and space costs on COL; (**d**) Construction time and space costs on NW.

Then, we conducted comparative experiments for V-Tree, G*-Tree, SILC, and the DVTG-Index.

As shown in Figure 10, it can be seen from the results in the figure that the construction time for V-Tree is significantly longer than that of the other indexes. This is because V-Tree must perform the Floyd algorithm for all subgraphs to calculate the distance matrix. Although the subgraphs in the upper layer only save the boundary points of the lower layer, the number of calculations is still very high. However, the DVTG-Index iteratively merges some useless vertex subgraphs upwards, reducing the number of boundary points and greatly optimizing the index construction. In addition, with the increase in the road network, the advantages of our method are more obvious.

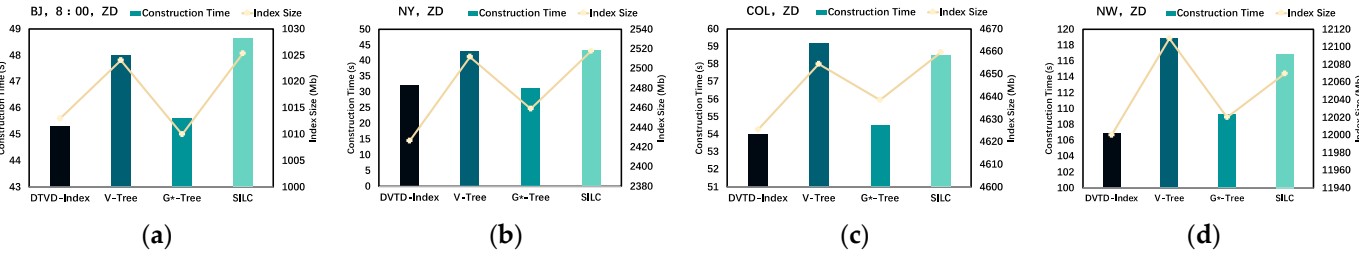

**Figure 10.** Comparison of construction time and space costs between baseline algorithms and DVTG-Index. (**a**) Construction time and space costs on BJ; (**b**) Construction time and space costs on NY; (**c**) Construction time and space costs on COL; (**d**) Construction time and space costs on NW.

The real-time change in moving objects is one of the difficulties in this study. Our index associated the moving objects with the vertices of the road network index and mapped the changes in the moving objects to the update of the index, which better handled this difficulty. When the moving object started to move on the road network, the index update performance also directly affected the performance of subsequent *k*-nearest neighbor queries. The number of moving objects and the index grid threshold affect the update performance of the index. Therefore, in this section, we evaluated the impact of various parameters on the performance of the DVTG-Index update.

First, we evaluated the impact of the number of moving objects on the index update time. In the BJ dataset, select 10%, 30%, 50%, 70%, and 100% of all taxis as the number of moving objects. In datasets NY, COL, and NW, 0.5%, 1.5%, 2.5%, 3.5%, and 5% of the number of vertices are generated as the number of moving objects, respectively. Among them, the index grid threshold $\eta$ is set to 8.

As shown in Figure 11, it can be seen from the experimental results that as the number of moving objects increases, the updated overhead time also increases. The result is consistent with the principle of our design; the increase in the number of moving objects leads to the more frequent splitting and merging of the index grid during the update process, and the calculation time of the vertex distance matrix also increases accordingly.

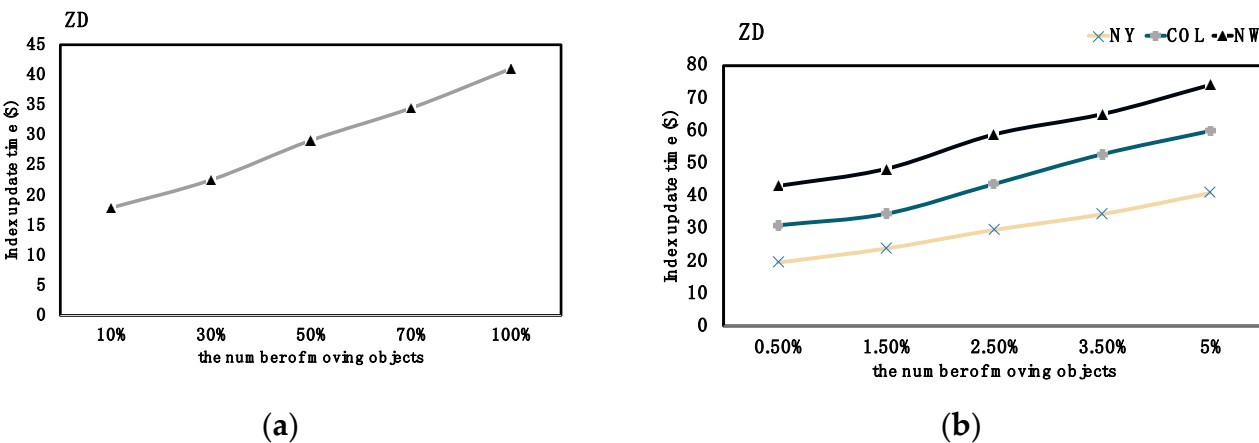

**Figure 11.** Time cost of updating DVTG-Index w.r.t. the number of moving objects. (**a**) Time cost on BJ; (**b**) Time cost on NY, COL and NW.

Then, we evaluated the impact of index grid threshold $\eta$ on the index update time. Among them, in the BJ dataset, the taxi trajectory between 8:00–9:00 on February 2 was obtained, and in datasets NY, COL, and NW, 5% of the number of vertices was used as the number of moving objects.

As shown in Figure 12, it can be seen from the experimental results that there is no positive or negative correlation between the index update time cost and index grid threshold $\eta$. When the threshold is relatively small, as the moving objects change, the index grid is more likely to be split down or merged up until the number of moving objects in the grid meets the threshold range, which takes a long time to update the index. As the threshold gradually increases, the number of moving objects that can be accommodated in the grid increases, and the time for grid scaling decreases so the total updated overhead time decreases. However, when the threshold gradually increases to a certain value, the number of active vertices in a single index grid is also very high, which makes the calculation time of the vertex distance matrix in the grid increase sharply. This makes the total time cost of the index update gradually increase with the increase in the threshold. In addition, the results of different datasets show that there are corresponding optimal thresholds in different road networks to minimize the update time cost, and the optimal value of the threshold increases with the increase in the dataset.

Finally, we conducted comparative experiments on the index update performance of V-Tree, G*-Tree, SILC, and the DVTG-Index. Dataset BJ takes 100% of taxis as moving objects, 5% of the number of generated vertices in datasets NY, COL, and NW as moving objects, and ZD as the moving-object distribution strategy.

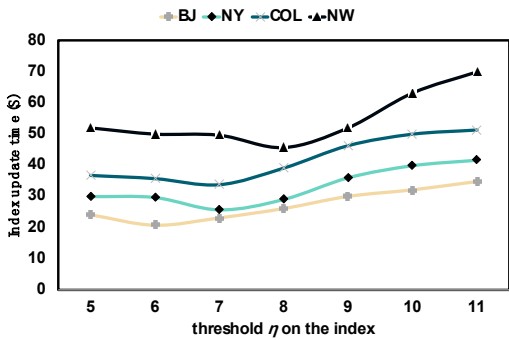

**Figure 12.** Index update time of DVTG-Index w.r.t. $\eta$.

As depicted in Figure 13, the empirical results demonstrate that the DVTG-Index exhibits the lowest and a relatively stable update time cost. This observed performance improvement of the DVTG-Index over benchmark algorithms is attributed to the reduction in the number of updated objects by associating moving objects with active vertices, thereby effectively reducing maintenance costs. As the size of the road network dataset increases, the comparative advantage of the DVTG-Index becomes increasingly pronounced. Conversely, the update of V-Tree and SILC requires a high time cost because, in the update process, they need to frequently traverse all the boundary points in the subgraph to search for the nearest interest point of the vertex.

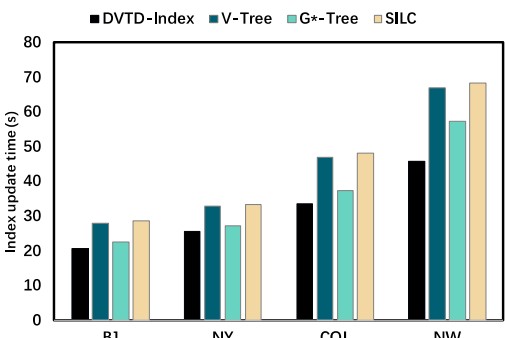

**Figure 13.** Comparison of update time between baseline algorithms and DVTG-Index.

### 6.4. Experimental Situation of the DVTG-CkNN Algorithm

The main aim of this paper was to solve the C$k$NN query problem; therefore, this section is the most important part of our experiment. To evaluate the C$k$NN query performance of our model, we compared our model with state-of-the-art methods. We evaluated our algorithm by varying the value of $k$, the number of moving objects, the strategy of moving-object distributions, and the number of consecutive queries.

First, we evaluated the effect of varying the value of $k$ on the performance of our algorithm and other benchmark algorithms for $k$NN queries for datasets BJ, NY, COL, and NW. We changed the value of $k$ to 1, 3, 5, 10, and 20, respectively, and recorded the overhead time required for related queries. Other experimental parameters presented continuous queries as 3, 10% of all taxis as moving objects in the BJ dataset, 2.5% of the number of vertices as the number of moving objects in the NY, COL and NW datasets, and the moving objects used the ZD distribution strategy.

As shown in Figure 14, obviously, as the value of $k$ increases, the query times of all algorithms increase; the average continuous query time of DVTG-C$k$NN is the shortest among the three algorithms. Taking the BJ dataset as an example, for $k$ = 5, the average single query time of DVTG-C$k$NN is about 79 μs, while the average single query time based on the G*-Tree is about 259 μs and the average query time based on V-Tree is about 1132 μs. It can be observed that the larger the value of $k$, the better the performance effect of the DVTG-C$k$NN algorithm than other algorithms. This is because the number of moving

objects in our index subgrid can be kept within a dynamic balance range, which reduces the complexity required to expand the search range and shortens the algorithm query time.

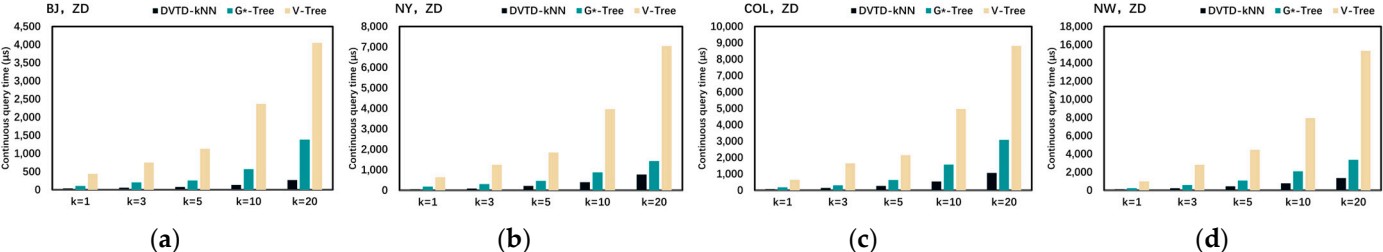

**Figure 14.** Comparison of query times between baseline algorithms and DVTG-C$k$NN w.r.t. $k$. (**a**) Query times on BJ; (**b**) Query times on NY; (**c**) Query times on COL; (**d**) Query times on NW.

Then, we evaluated the impact of the number of moving objects on the performance of $k$NN queries on BJ and COL datasets. In the BJ dataset, we selected 10%, 30%, 50%, 70%, and 100% of all taxis as moving objects, and in the COL dataset, we, respectively, generated 0.5%, 1.5%, 2.5%, 3.5%, and 5% of the number of vertices as the number of moving objects and set other experimental parameters $k = 5$, and the distribution strategy of moving objects adopted the ZD distribution.

As shown in Figure 15, we can draw the following conclusions. Firstly, as the number of moving objects increases, the average query overhead time of the three algorithms decreases. This is because the increase in the number of moving objects increases the density of the moving objects in the search grid and reduces the range to be searched. Secondly, the average continuous query time of DVTG-C$k$NN is the shortest among the three algorithms; even in the case of the high number of moving objects, DVTG-C$k$NN still has a great advantage. As previously designed, the DVTG-C$k$NN algorithm is based on the DVTG-Index, which divides the large road network into grids and dynamically merges or splits them according to the number of active vertices in the grid. This keeps the moving objects in different grids in a relatively balanced range, which can significantly improve the efficiency of querying moving objects in large road networks.

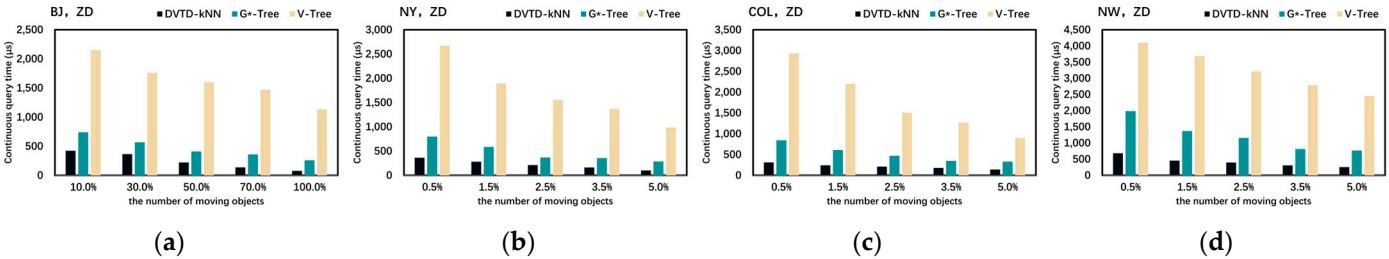

**Figure 15.** Comparison of query times between baseline algorithms and DVTG-C$k$NN w.r.t. the number of moving objects. (**a**) Query times on BJ; (**b**) Query times on NY; (**c**) Query times on COL; (**d**) Query times on NW.

Then, we evaluated the impact of different distribution strategies of moving objects on the performance of the $k$NN query using three distribution strategies to generate the number of moving objects for datasets BJ and NW, respectively. Experiment with other parameters $k = 5$, and the number of moving objects is 2.5% of the number of vertices in the dataset.

As shown in Figure 16, we can draw the following conclusions. First, under the three distribution strategies of RD, ND, and ZD, the overhead times of the three query algorithms increases in turn. The uneven distribution of moving objects under the three distribution strategies gradually increases, and the search range gradually increases, requiring more overhead time. Secondly, the average single query time and average continuous query time of DVTG-C$k$NN are the shortest among the three algorithms. Especially under the ZD

distribution strategy, the performance advantage of DVTG-C*k*NN is more obvious than that of the benchmark algorithm. This is because the DVTG-C*k*NN algorithm is based on the DVTG-Index, which is designed for unevenly distributed dynamic road networks, and balances the uneven distribution of moving objects by dynamically dividing the grid.

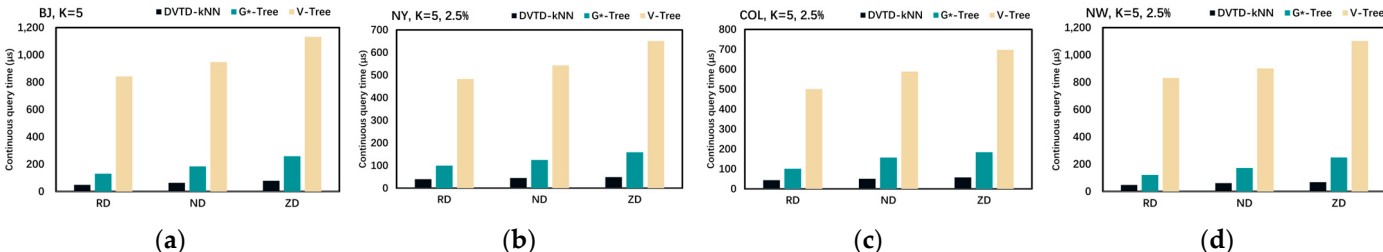

**Figure 16.** Comparison of query times between baseline algorithms and DVTG-C*k*NN under the three distribution strategies. (**a**) Query times on BJ; (**b**) Query times on NY; (**c**) Query times on COL; (**d**) Query times on NW.

Finally, we evaluated the impact of different consecutive query times on the performance of *k*NN queries. We set the number of continuous queries as 1, 3, 5, 10, and 20, and the moving objects used the ZD distribution strategy. Other experimental parameters were *k* = 5, 10% of all taxis as moving objects in the BJ dataset, and 2.5% of the number of vertices as the number of moving objects in the NY, COL, and NW datasets, and the moving objects used the ZD distribution strategy.

As shown in Figure 17, the average continuous query time of DVTG-C*k*NN is the shortest among the three algorithms. Especially after the number of continuous queries gradually increases, the performance advantage of DVTG-C*k*NN compared with the baseline algorithm is more obvious. This is because the DVTG-C*k*NN algorithm provides traversal for subsequent continuous queries through the query vertex queue. The boundary vertices in the queue are not updated as the position of the moving object changes. Therefore, the border vertex data in the queue can still be used again in continuous queries, and the number of border vertices is greater than that of active vertices; therefore, the repeated traversal of vertices can be reduced.

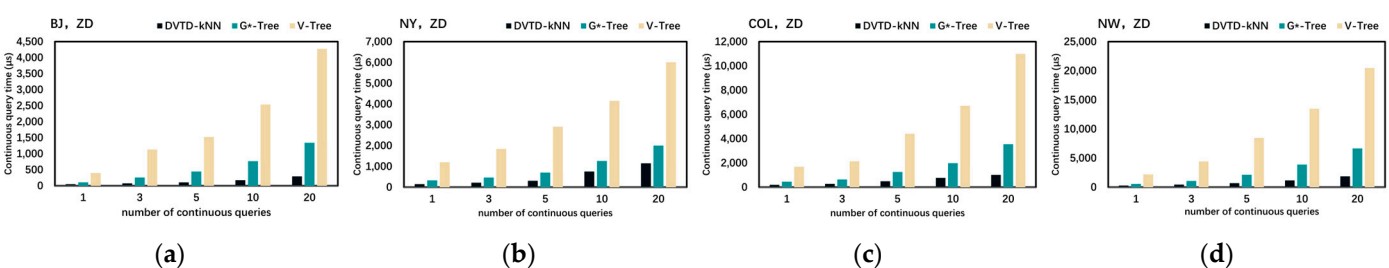

**Figure 17.** Comparison of query times between baseline algorithms and DVTG-C*k*NN w.r.t. the number of continuous queries. (**a**) Query times on BJ; (**b**) Query times on NY; (**c**) Query times on COL; (**d**) Query times on NW.

## 7. Conclusions

This paper studied the problem of the continuous *k*-nearest neighbor query for moving objects in a road network, and proposed the DVTG-Index, a dynamic multi-layer grid index, which can index moving objects in the road network and update them in real time. Secondly, this paper provided a continuous *k*-nearest neighbor query method for moving objects in a road network based on the DVTG-Index. DVTG-C*k*NN can perform efficient and continuous *k*-nearest neighbor queries, and the query efficiency reaches the microsecond level, which is obviously better than the mainstream algorithm used at present for solving this problem. Finally, we conducted experiments on a real road network dataset. The experimental

results show that the DVTG-Index and DVTG-C$k$NN proposed in this chapter present better performance. Especially in the face of large road network maps and uneven distributions of moving objects, it has more rapid response capabilities. We also know that our method presented limitations and deficiencies. The $k$-nearest neighbor query method proposed in this paper is based on the optimal condition of single query processing. However, when the actual platform processed query requests, the sum of all individual optimal query processes may not be the optimal processing method for the overall situation. In the following stage of the research, we will focus on the $k$-nearest neighbor query algorithm with globally optimal query processing methods, which is of great significance to improve the responsiveness of the platform, and it also provides a good direction for the follow-up research. At the same time, in real life, road conditions must be considered. The weight value of the road network should not only be the length of the road, but should also be added to the road traffic conditions, and the road traffic cost should be considered as the weight value. Therefore, the future work can consider optimizing the road network model and enhancing the practicability of the road network-based $k$-nearest neighbor query algorithm.

**Author Contributions:** Conceptualization, K.T. and Z.D.; Methodology, K.T.; Software, K.T.; Validation, K.T., Z.D. and W.S.; Formal analysis, K.T.; Investigation, K.T.; Resources, Z.D.; Data curation, K.T.; Writing—original draft, K.T.; Writing—review & editing, Z.D.; Visualization, K.T.; Supervision, Z.G.; Project administration, Z.G.; Funding acquisition, Z.G. All authors have read and agreed to the published version of the manuscript.

**Funding:** This research was funded by Shanghai Municipal Science and Technology Major Project (No. 2021SHZDZX0103), the Science and Technology Commission of Shanghai Municipality (No. 19511132000), and the Department of Science and Technology of Guangdong Province (Grant No. 2019A1515110352).

**Institutional Review Board Statement:** Not applicable.

**Informed Consent Statement:** Not applicable.

**Data Availability Statement:** Not applicable.

**Conflicts of Interest:** The authors declare no conflict of interest.

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
