# Peer review of "A Dynamic Grid Index for CkNN Queries on Large-Scale Road Networks with Moving Objects"

_applsci, doi:10.3390/app13084946_

Round 1

Reviewer 1 Report

I believe that the paper overall is a good paper but I suggest that after related work section and before the section of 3. Dynamic V-Tree Double-Layer Grid Index  to put a section explain how the next sections go and the methods of the work.

Reviewer 2 Report

The authors have designed a DVTG-CkNN algorithm to speed up the CkNN query on large-scale road networks in the real-time scenario, which enhances the novelty of their work. However, the real-time implementation also raises various concerns, which are mentioned as follows:

1) The authors need to improve the related work section by defining the identified research gaps of the existing literature, which can prove the novelty and benefit of their proposed work over literature. Moreover, how authors compare the proposed system with the existing literature without any comparative analysis table, which will also give clarity about the research gaps or issues associated with the literature work.

2) In subsection 4.2, the authors define some of the concepts for explaining the query algorithm with the help of Definition 13, Definition 14, and Definition 15. But the theoretical concepts are not defined with clarity. It can be further improved and enhanced with the help  of mathematical proof or some mathematical formulation for the same.

3) Section 4.2 discusses the time complexity of the DVTG-CkNN algorithm which does not seem to be an accurate result in case of the large number of active, border, and adjacent vertices. Then, how authors are going to analyze the time complexity of the proposed scheme, which also increases the computation time of the proposed work, further deteriorating the performance of the system.

4) Moreover, the authors have not analyzed the space complexity of the proposed DVTG-CkNN algorithm which can impact the efficiency and reliability of the proposed algorithm. As the requirement of huge storage space in case of a large number of active, adjacent, and border vertices can affect the performance of the system and it can also incur huge data storage costs if we don't analyze the space complexity of the proposed algorithm.

5) The proposed k-nearest neighbor algorithm in this paper may not yield the optimal solution for moving objects in the road network. As the designed algorithm is not applicable to large road networks and divergent moving objects making it unresponsive to the platform or network. So, authors should redesign and upgrade their algorithm to make it applicable and feasible for large or uneven road networks; otherwise, it detriments the feasibility and efficiency of the network for large road networks.

6) The paper contains many grammatical mistakes and it is difficult to identify many abbreviations considered in the paper. So, authors should thoroughly proofread the paper to check the vocubulary and prepare a table defining all the terms used in the paper.

Reviewer 3 Report

Abstract: Abstract needs to be improved following problem under study, research methods, results, conclusion and significance.

Introduction: It does not clearly state the main problem that the paper aims to solve. Although it mentions the importance of location-based services and the challenges involved in the kNN problem in the road network environment, it does not provide a clear problem statement until the end of the introduction. This lack of clarity could make it difficult for readers to understand the significance of the proposed solution and its potential impact. Additionally, the introduction could benefit from a more concise and focused presentation of the background information and related work to avoid overwhelming the reader with too many details.

Related Works: It only briefly mentions the challenges and limitations of existing methods for continuous k-nearest neighbor (kNN) queries involving moving objects, without providing a detailed analysis or critical evaluation of these methods. Additionally, the section does not clearly state the specific research gap or problem that the proposed research aims to address. Furthermore, the section lacks clarity and organization, making it difficult for the reader to understand the various proposed index structures and their respective strengths and weaknesses.

Weakness of the DVTG-CkNN query algorithm is that it may not perform optimally in highly dense areas with a large number of continuous queries. The algorithm was designed to improve efficiency by saving the search range of the initial query and performing incremental calculations for subsequent queries. However, in areas with a high density of queries, the number of saved search ranges may become too large, leading to a reduction in efficiency. Additionally, the algorithm may become less effective as the number of moving objects increases, as the number of incremental calculations needed to update the kNN of each query request increases. Therefore, the algorithm may require further optimization to handle highly dense areas and a large number of moving objects.

DVTD-kNN algorithm is its time complexity, which is difficult to accurately evaluate due to its dependence on the number of active and boundary vertices near the query point and their relationships with each other. The time complexity of the algorithm can be assumed to be O(k) in the best case scenario where the number of active vertices is smaller than the number of boundary vertices, and the number of adjacent vertices of each boundary vertex is also small. However, in the worst case scenario where the number of active vertices is large and the number of border vertices adjacent vertices is also large, the time complexity of the algorithm may be high. Moreover, the time complexity of the algorithm also depends on the value of k, where a large value of k can result in higher time complexity. Therefore, choosing an appropriate value of k is crucial for optimizing the performance of the algorithm.

Experimental Evaluation:  The experiments only evaluate the performance of the proposed algorithm (DVTG-kNN) against two benchmark algorithms (G*-Tree and V-Tree) on two specific datasets (BJ and NY). While this provides some insight into the performance of the algorithm, it is not clear how well it would perform on other datasets or against other state-of-the-art algorithms.

The text provides limited details about the experimental setup, such as the specific hardware used, the size of the datasets, and the number of queries executed. Without this information, it is difficult to fully evaluate the results or to reproduce the experiments.

The text primarily focuses on presenting the results of the experiments without providing a detailed analysis or interpretation of the results. It is not clear, for example, why the proposed algorithm performs better than the benchmark algorithms or what factors might be driving the observed performance differences.

The text does not compare the proposed approach to other recent approaches that solve the same problem. This comparison could help readers understand the state of the art and how the proposed approach compares to other recently developed solutions.

The text does not discuss the limitations of the proposed approach, such as the conditions under which it might not perform well or the types of datasets for which it may not be well-suited. Such limitations are important for readers to understand in order to evaluate the practical usefulness of the approach.

Conclusion: Lack of discussion on the limitations of the proposed DVTG-Index and DVTG-CkNN methods. While the experimental results showed better performance, it is important to acknowledge any limitations or drawbacks of the proposed methods. This could have provided a more comprehensive and balanced conclusion for the study.

Abbreviations should be defined at first mention and used consistently thereafter.

References should be used from resent years and the need for extensive literature review is required so that the comparison form other works can be reflected in the article.

Round 2

Reviewer 2 Report

The authors have addressed all the mentioned comments. Now the paper is in good shape. 

Author Response

Thanks for your letter and reviewers’ professional comments concerning our manuscript entitled “A Dynamic Grid Index for CkNN Queries on Large-Scale Road Networks with Moving Objects”. The reviewer’s comments are valuable and helpful for revising and improving our paper. We have revised the manuscript according to your detailed and kind advice and reviewers’ suggestions. Revised portions are highlighted in blue in the paper. The main corrections in the paper and response to the reviewers’ comments are as followed:

Point 1: The authors have addressed all the mentioned comments. Now the paper is in good shape.

Response 1: Thank you very much for your feedback and guidance throughout the review process. We are pleased to hear that our revisions have addressed all of the reviewers' concerns and that the paper is now in good shape. We appreciate the time and effort that you and the reviewers have put into evaluating our work. We have taken your comments seriously and made every effort to improve the quality of the article, including improving the detailed expression of the article. We believe that these revisions have strengthened the paper and hope that it will be accepted for publication. Thank you again for your valuable feedback.

We have tried our best to improve the manuscript and made changes for better understanding. Special thanks for all valuable comments and suggestions.

Yours sincerely

Kailei Tang

Reviewer 3 Report

While the abstract provides a good overview of the paper, there is room for improvement in terms of providing more specific details on the research problem, proposed solution, and research findings.

I am happy with the introduction.

I am happy with the related works.

I am happy with the methodology.

I am happy with the DVTG-CkNN query algorithm.

I am happy with the experiment analysis.

I am happy with the conclusion.

Still abbreviations are not addressed properly e.g. G-Tree should be grid tree and so on within the article.

Kindly fix these minor issue inorder to be ready for publication.
